

# How reliable is metabarcoding for pollen identification? An evaluation of different taxonomic assignment strategies by cross-validation

Gilles San Martin[1], Louis Hautier[1], Dominique Mingeot[2] and
Benjamin Dubois[2]

[1] Life Sciences Department, Plant and Forest Health Unit, Walloon Agricultural Research Centre,
Gembloux, Belgium
[2] Life Sciences Department, Bioengineering Unit, Walloon Agricultural Research Centre,
Gembloux, Belgium

Corresponding author
Gilles San Martin,
g.sanmartin@cra.wallonie.be

## ABSTRACT

Metabarcoding is a powerful tool, increasingly used in many disciplines of environmental sciences. However, to assign a taxon to a DNA sequence, bioinformaticians need to choose between different strategies or parameter values and these choices sometimes seem rather arbitrary. In this work, we present a case study on ITS2 and *rbcL* databases used to identify pollen collected by bees in Belgium. We blasted a random sample of sequences from the reference database against the remainder of the database using different strategies and compared the known taxonomy with the predicted one. This *in silico* cross-validation (CV) approach proved to be an easy yet powerful way to (1) assess the relative accuracy of taxonomic predictions, (2) define rules to discard dubious taxonomic assignments and (3) provide a more objective basis to choose the best strategy. We obtained the best results with the best blast hit (best bit score) rather than by selecting the majority taxon from the top 10 hits. The predictions were further improved by favouring the most frequent taxon among those with tied best bit scores. We obtained better results with databases containing the full sequences available on NCBI rather than restricting the sequences to the region amplified by the primers chosen in our study. Leaked CV showed that when the true sequence is present in the database, blast might still struggle to match the right taxon at the species level, particularly with *rbcL*. Classical 10-fold CV—where the true sequence is removed from the database—offers a different yet more realistic view of the true error rates. Taxonomic predictions with this approach worked well up to the genus level, particularly for ITS2 (5–7% of errors). Using a database containing only the local flora of Belgium did not improve the predictions up to the genus level for local species and made them worse for foreign species. At the species level, using a database containing exclusively local species improved the predictions for local species by ~12% but the error rate remained rather high: 25% for ITS2 and 42% for *rbcL*. Foreign species performed worse even when using a world database (59–79% of errors). We used classification trees and GLMs to model the % of errors *vs.* identity and consensus scores and determine appropriate thresholds below which the taxonomic assignment should be discarded. This resulted in a significant reduction in prediction errors, but at the cost

of a much higher proportion of unassigned sequences. Despite this stringent filtering, at least 1/5 sequences deemed suitable for species-level identification ultimately proved to be misidentified. An examination of the variability in prediction accuracy between plant families showed that *rbcL* outperformed ITS2 for only two of the 27 families examined, and that the % correct species-level assignments were much better for some families (*e.g.* 95% for Sapindaceae) than for others (*e.g.* 35% for Salicaceae).

# INTRODUCTION

Metabarcoding has been increasingly used during the last 10 years in various fields of ecology and environmental sciences to identify plants and animals (*Taberlet et al., 2012*; *Deiner et al., 2017*; *Ruppert, Kline & Rahman, 2019*; *Liu et al., 2020*; *Bell et al., 2022*). Metabarcoding has particularly attracted scientists' interest as a tool to identify pollen collected by pollinators, which has many applications, such as understanding plant-pollinator networks, assessing the impact of foraged plants on the nutritional quality of their food, tracing back the source of contamination by pesticides and other toxic products, *etc*. The traditional microscopic identification approaches have several drawbacks: they require highly skilled specialists, the process is slow and their taxonomic resolution is generally poor (up to the family level, genus at best). Metabarcoding could overcome these issues and has the potential to revolutionise this field (*Bell et al., 2016*, *2022*).

However, in order to assign a DNA sequence to a taxon, bioinformaticians have to choose between different strategies and parameter values. It is essential to assess how these choices affect the reliability of identifications.

Firstly, different DNA barcode sequences can be used. For vascular plants and pollen in particular, the most frequently used are ITS2, *rbcL*, *matK*, *trnL* and ITS1 (*Burgess et al., 2011*; *Richardson et al., 2015*; *Braukmann et al., 2017*; *Utzeri et al., 2018*; *Kamo et al., 2018*; *Voulgari-Kokota et al., 2019*; *Bell et al., 2019*, *2020*, *2021*; *Gous et al., 2019*; *Potter et al., 2019*; *Tremblay et al., 2019*; *Bänsch et al., 2020*; *Baksay et al., 2020*; *Swenson & Gemeinholzer, 2021*; *Coghlan, Shafer & Freeland, 2021*; *Wilson et al., 2021*; *Milla et al., 2021*; *De Jesus Inacio et al., 2021*; *Jones et al., 2021*; *Arstingstall et al., 2021*; *Quaresma et al., 2021*).

Besides the choice of barcodes, another strategic choice concerns the sequences to include in the reference database. For example, some publications limit the reference database to the local fauna or flora living in the study area (*e.g. Milla et al., 2022*). While this might seem like a wise approach, it remains uncertain whether it will consistently enhance the accuracy of taxonomic assignments. Indeed, if the local flora/fauna is poorly represented in the database, using the species from a wider geographic range might improve the taxonomic assignments, at least at higher taxonomic levels than the species.

Sometimes, detecting exotic species might also be of particular interest (*e.g.* to determine which non-native plants are foraged by bees in urban environments).

Moreover, the reference databases contain sequences of various lengths. In metabarcoding studies, the query sequences are rather short, depending on the sequencing technology (max. 550 bp with the most commonly used Illumina MiSeq platform) and the primers used. Some authors (*e.g. Richardson et al., 2021*) chose to use reference databases with sequences restricted to the amplified portion, while other studies relied on the whole sequences (*e.g. Bell, Loeffler & Brosi, 2017*). The reference database can also be dereplicated in different ways: *e.g.* a "majority mode" might attribute the duplicated sequences to the most frequent taxon while a "unique mode" would keep all taxa sharing the same sequence (*Dubois et al., 2022*).

Next, bioinformaticians need to select one or several taxonomic assignment algorithms, choose various parameter thresholds and design a strategy to evaluate the taxonomic level at which the identification is reliable. Blast+ (*Camacho et al., 2009*) is one of the most widely used classification algorithms, at least for pollen metabarcoding. This algorithm is indeed relatively fast, it provides easy-to-interpret outputs and it can perform very well compared to other algorithms in terms of taxonomic assignment accuracy (*Braukmann et al., 2017*; *Bokulich et al., 2018*; *Hleap et al., 2021*). However, even if one chooses to use blast, there remain several ways to assign a taxonomy, *e.g.* according to the highest bit score or the most frequent taxon among the top N hits, with or without filtering out the results that do not meet certain criteria (*e.g.* minimum 50% of consensus and minimum 97% of identity at the species level).

In the literature, the reasons behind the diverse choices in the metabarcoding process or the selected threshold values for different parameters are often poorly documented or lack justification. Many authors seem to simply use the default parameters of the chosen pipeline or algorithm. However, choosing the right parameter values is of tremendous importance to optimise the taxonomic assignment results (*Bokulich et al., 2018*; *Hleap et al., 2021*).

To assess the impact of these methodological choices on identification accuracy, one needs to test the method in cases where the true taxonomic identity of the target sequences is known. There are three main ways to do this, each with different advantages and limitations: biological samples, mock communities, and cross-validation (*Bokulich et al., 2020*).

Many studies compare metabarcoding of pollen samples with classical microscopic identification, but this approach suffers from uncertainty about the true taxonomic composition, particularly at lower taxonomic level (*Richardson et al., 2015*, *2021*; *Smart et al., 2017*; *Leontidou et al., 2018*; *Macgregor et al., 2019*; *Richardson et al., 2019*; *Bänsch et al., 2020*). Mock communities offer many advantages as they are as close as possible to biological samples but with known composition. However, they are limited in the number of species combinations that can be tested and they can be technically challenging due to issues like environmental contamination and source material availability (*Hleap et al., 2021*). While more artificial than mock communities, cross-validation is a purely computational method that allows testing many taxa and strategies at minimal cost.

It involves extracting random sequences from a reference database and predicting their taxonomy using different pipelines and parameter choices, which are then compared to the "true" taxonomy from the original database.

The aim of this article is to focus on the possibilities offered by cross-validation using a study case: ITS2 and *rbcL* databases used for metabarcoding identification of pollen collected by bees in Belgium with blast as a classification algorithm. This approach not only offers valuable insights for researchers in the field of plant-pollinator interactions but also allows us to explore in depth what can be done with this simple *in silico* methodology. The general approach presented here remains however applicable to any reference database of identified DNA sequences.

Several questions, grouped in three categories, are investigated through this cross-validation approach:

(I) Database characteristics: what are the strengths and weaknesses of the two chosen barcodes (ITS2 and *rbcL*)? Is it better to work with a database containing only the local flora or to work with a larger reference database including all species available from around the world? What is the influence of using shorter reference sequences restricted to the amplified portion of the gene *vs.* whole sequences?

(II) Cross-validation approach: since the target sequences can either be removed (10-fold CV) or not (leaked CV) from the reference database before blasting, what is the influence of these two approaches on the results and how to interpret the differences?

(III) Taxonomic assignment strategies: What is the best way to assign the taxonomy with blast (best hit, majority vote among the 10 best hits, *etc.*)? Up to which taxonomic level can we reasonably expect a reliable identification? How can we use simple descriptive statistics like the identity or consensus scores to determine how reliable an identification is? How much does the identification accuracy vary between different plant families?

## MATERIALS AND METHODS

Our general approach was to extract a random sample of sequences from existing ITS2 and *rbcL* barcodes databases, blast them against the remainder of the database and compare the predicted taxonomies with the "true" taxonomy associated with each sequence. This approach allows to evaluate the impact of different strategies (detailed below) of taxonomic assignment and database building on the accuracy of the predicted taxonomies.

### Various strategies for taxonomic assignment, database choices and cross-validation approaches

#### Barcodes: rbcL vs. ITS2

We chose to work with databases dedicated to the ITS2 and *rbcL* barcodes because they are commonly used and recommended for plant/pollen metabarcoding (*e.g.*: *Bell et al., 2022*; *Lowe et al., 2022*). All available sequences were retrieved from the National Center for Biotechnology Information (NCBI) and cleaned with the DB4Q2 pipeline (*Dubois et al., 2022*). In summary, sequence and taxonomy data was retrieved from NCBI and reformatted, low-quality sequences were discarded, a step of dereplication was carried out to reduce redundancy, two additional filters were applied to remove sequences suspected to

have a fungal origin or to be misidentified, and a last optional step consisted in the database restriction, *i.e.* extracting from reference sequences only the portion amplified by a primer set of interest. The resulting databases cover a minimum of 81% (ITS2) to 87% (*rbcL*) of the species present in the wild in our study case area (Belgium) and approximately 95% of the genus.

### Region of the barcode gene considered: General, Restricted, Restricted-General

Since the reference sequences were downloaded from NCBI without a stringent length filter, they displayed variable lengths: inter-quartile range between 547 and 688 pb for ITS2 (min = 109 pb, max = 14,531 pb) and between 551 and 1,331 pb for *rbcL* (min = 100 pb, max = 2,550 pb). However, with a metabarcoding approach, only a fraction of these sequences is amplified depending on the chosen primers. For example, the primer sets ITS-S2F (5′-ATGCGATACTTGGTGTGAAT-3′)/ITS4R (5′-TCCTCCGCTTATTGATATGC-3′) and rbcLaF (5′-ATGTCACCACAAACAGAGACTAAAGC-3′)/rbcLr506 (5′-AGGGGACGACCATACTTGTTCA-3′) that have been used in our case study, amplify fragments with median lengths of 343 and 458 pb respectively, with little variation.

Using the General DB containing the full-length sequences could artificially increase the accuracy of the taxonomic predictions. Indeed, with longer sequences more information is available to distinguish closely related species, whereas with a metabarcoding approach the typical amplicon length is rather short. Alternative databases were therefore built for each barcode restricted to the region amplified by the ITS-S2F/ITS4R and rbcLaF/rbcLr506primers (Restricted DB). The restricted databases contained, however, a much lower number of sequences: ∼22,000 and ∼15,000 sequences for ITS2 and *rbcL* respectively while the general databases contained ∼170,000 and ∼135,000 sequences respectively. Also, the restricted DB were not an exact subsample of the general DB because the clipping step appears before the cleaning step in our pipeline. In some cases, the full sequence was discarded because it did not pass the quality checks, while the sequence restricted to the amplified region passed the quality check and was conserved in the database. The drop in the number of sequences can be explained by two main reasons: (i) the full-length sequences only partially overlap the target region and/or (ii) the primer hybridization site of the full-length sequence has too many SNPs compared to the primer sequences provided for the database clipping.

In these two approaches, the Restricted or General DB were cross-validated against themselves. A third approach, "Restricted-General", was also tested, in which the cross-validation involved blasting the restricted sequences against the general databases.

### Species origin for the target sequence: Foreign sp. vs. Local sp.

To evaluate how the taxonomic assignment performs for species of different origin, the target sequences were classified into two categories: Local species *vs*. Foreign species. The Local sp. category is considering the flora susceptible to be found in our area of interest (Belgium) in a wide sense (∼2,000 species): it contains native species but also crops and non-native plants present in the wild. All other species from the the database were

considered as "Foreign species" even if they might be planted in urbanized areas. These exotic plants can, however, be an important source of pollen for bees (*Casanelles-Abella et al., 2022*) and are of particular interest in our study case.

### Database geographic area: Local DB vs. World DB

Reference databases containing only the local species (Local DB) were created, to compare with the result obtained when all sequences were kept in the database, *i.e.* corresponding to species from the entire world (World DB). We can then blast Local sp. sequences against the Local DB or World DB and compare the quality of taxonomic assignments obtained when we blast Foreign sp. against the same databases.

To be clear: the Local species and Foreign species labels concern the sequences that are being blasted, while Local DB and World DB concern the databases against which the sequences are blasted. The Local DB contains all the sequences of the Local sp. while the World DB contains the Local sp. + Foreign sp. sequences.

### Cross-validation strategy: 10-fold CV vs. leaked CV

We tested two different ways to perform the cross-validation (CV). The 10-fold CV is the most conventional approach: the sequences are divided into 10 groups (=folds), each containing a random sample of 10% of the sequences. The sequences from each fold are then blasted against the remaining 90% of the sequences from the other folds. With the other approach named here (leaked CV), the sequences were blasted against the full database without excluding the target sequences. These two approaches are testing different hypotheses. With leaked CV, we test the ability of blast to match the exact sequence when it is present in the database among many other sequences. With 10-fold CV we test how blast performs to find alternative sequences that are taxonomically close to the true target sequence.

Also, we did not perform a full cross-validation for all the sequences. We kept all sequences corresponding to Local species (~32,000 sequences across ITS2 and *rbcL* databases) as well as a random sample of 10% of the sequences corresponding to "Foreign species" from each database (~25,000 sequences across ITS2 and *rbcL* databases). We also excluded the sequences with imprecise identifications (*i.e.* not identified to the species level), doubtful names and hybrids taxa. In the end, ~31,000 ITS2 and ~25,000 *rbcL* sequences were blasted against the general databases and ~4,800 ITS2 and ~3,500 *rbcL* sequences against the restricted databases.

For the Restricted *vs.* general 10-fold CV approach, the restricted sequences were split into 10 folds containing 10% of the target restricted sequences. Before blasting each fold against the general database, the full-length sequences corresponding to the restricted sequences present in the fold were removed from the reference database.

### Descriptive statistics

#### Identity and consensus scores

From each blast on a target sequence, we extracted the 10 best hits (based on the bit score) and computed two types of descriptive statistics. The bit score provided by blast combines the identity score and the length of the alignment. It can be used to rank different matches

for a given target sequence but it cannot be used to compare the quality of the match between different target sequences. The **identity score** (%), directly provided by blast, gives the percentage of base pairs (bp) that are identical between the target sequence and the sequence matched in the reference database. Once a taxon is assigned, we attribute the identity score corresponding to the maximum bit score for this taxon. NB: this is not necessarily the maximum identity score because the bit score penalises matches with low values for alignment length (*i.e.* it is better to have 98% of Identity with a 400 pb alignment length than to have 100% identity with a 50 pb alignment length. In this case, 98% would be the identity score assigned to the taxon). The **consensus score** is simply the percentage of the 10 top hits matching the same taxon. If among the top 10 hits, blast matches two *Brassica napus*, six *Brassica nigra* and two *Sinapis alba*, the consensus score is 60% for *Brassica nigra*, 80% for the genus *Brassica* and 100% for the family *Brassicaceae*.

*Quality of the taxonomic predictions*
Once a taxonomy is attributed to a target sequence, it can be compared with the "true" taxonomy at each taxonomic level (family, genus, species). When looking at all sequences together, the percentage of sequences that are correctly predicted can be computed (**Accuracy**). We considered here that when there is no taxon predicted, the sequence was not successfully predicted to obtain an estimate of the best accuracy possible. However, if the aim is to estimate the quality of predictions separately for various taxa, there are in fact several ways to do so. In order to estimate, for example, the % of correct predictions for the family *Brassicaceae*, we can simply compute the % of sequences that are predicted to be *Brassicaceae* and that indeed belong to this family. This metric is called the **Precision** (P = True Positive/(True Positive + False Positive)). Another option is to compute the % of "True Brassicaceae" that are correctly predicted. This metric is called the **Recall** or **Sensitivity** (R = True Positive/(True Positive + False Negative)). The **F-score** combines these two metrics by computing their harmonic mean: F = 2∗P∗R/(P+R).

Note that with a multi-class confusion matrix, the total true positive, total false positive and total false negative can be computed over all classes (here species, genus, families, *etc.*). In that case, the resulting global metrics (for all classes) are all equivalent: Accuracy = Recall = Precision = F-score (sometimes called "Micro F1") (*Scikit-learn Help, 2022*, sec. 3.3.2.9.2.). Another option, not used here, is to compute Precision, Recall and F-score for each species, genus, family and average them ("macro-averaging"). The scores obtained will be different and might over-emphasize the typically low performance of infrequent classes (*Scikit-learn Help, 2022*, sec. 3.3.2.1.). However we believe that for this particular application, accuracy (=Micro F1) is a straightforward and more intuitive way to express the global taxonomic prediction performance (*i.e.* we have ~30,000 sequences, how many are correctly predicted?).

**Blast taxonomic assignment methods: TopHit, TopHitPlus, TopN, TopNplus**
Four different methods were used to assign a taxon from the top 10 blast hits, two are based mainly on the highest bit score and the two others on the consensus scores. Both the identity and consensus scores are computed for each of the four methods and for each

taxonomic level because they can be useful to assess the quality of the taxonomic assignments.

i) With TopHit, the taxon with the best Bit score is selected (default blast output if we keep only one match).

ii) With TopHitPlus, the taxon with the best bit score is also retained, but the potential bit score ties are broken by choosing the taxon with the highest consensus score among the ties.

iii) With TopN, the taxon with the highest consensus score among the top 10 hits is selected at each taxonomic level. With this approach, the chosen species might belong to a different genus than the taxon chosen at the genus level for example.

iv) With TopNPlus, the sequences which do not fulfill some predefined requirements are first eliminated, then the consensus score is computed and the taxon with the highest consensus score is computed at each taxonomic level. In this study, we discarded the matched sequences with an alignment length < 100 base pairs and with an E score > $10^{-10}$. The identity score had also to be at least 97% for the species level, 90% for the genus level and 80% for the family level. Also the difference of identity score between the best hit and the matched sequence had to be <1% for the species level, 10% for the genus level and 20% for the family level (approach inspired by *e.g. Milla et al. (2022)* and https://github.com/Joseph7e/Assign-Taxonomy-with-BLAST).

Note that the rationale behind the three first methods is to assign a taxon at each taxonomic level (*e.g.* species, genus, family) and then to evaluate the reliability of the prediction in a second stage to choose a reasonable taxonomic level. These three methods always provide a predicted taxonomy, with the rare exception where blast finds no match (representing <2% of the sequences for particular cases). With TopNPlus, we must choose *a priori* a series of criteria that will prevent the algorithm from providing any prediction at a given taxonomic level if these criteria are not met. Sometimes none of the top 10 sequences matches the requirements. In that case, no taxon is attributed.

### Use of the identity and consensus scores to evaluate the reliability of the taxonomic assignments

Two types of algorithms were used to predict the quality of the taxonomic assignments: binomial generalized linear models (GLMs) and classification trees (based on the Gini index). In both cases, the response was a binary variable indicating whether the identification is correct or not. The predictors were the identity and consensus scores (plus their interaction for the GLMs). Separate models were fit at each taxonomic level and for each barcode. The classification trees were pruned with the one standard error rule on the cross-validation error rate. GLMs are particularly suited for modelling linear increases (on the logit scale here) and to visualise the correlations. Classification trees are particularly suited to find discontinuities and help to choose thresholds under which the identification should be considered as unreliable.

## Software

The reference databases were built using QIIME2 (*Bolyen et al., 2019*) and its RESCRIPt plugin (*Robeson et al., 2021*) with the DB4Q2 pipeline (*Dubois et al., 2022*). We used the BLAST+ command line application version 2.9.0+ to extract the top 10 hits from the reference databases (*Camacho et al., 2009*). The rest of the computations and taxonomic assignments were performed in the R programming language (*R Core Team, 2022*) with home made scripts. Classification trees were fit with the package rpart 4.1-15 (*Therneau & Atkinson, 2019*) and plotted with partykit 1.2-13 (*Hothorn & Zeileis, 2015*). Programs to perform database cross-validation already exist (*e.g.* the excellent RESCRIPt QIIME2 plugin, *Robeson et al. (2021)*). However we needed more flexibility to test various strategies and then to explore graphically the outputs. R combined with several tidyverse packages (*Wickham et al., 2019*) offers such an extremely flexible environment for both data manipulation and visualisation. Most of the key operations have been embedded in R functions and in a fully documented R package to make the code reusable and well structured (https://github.com/GillesSanMartin/CVrefDB). The raw data (databases), transformed data (assigned taxonomies and descriptive statistics) and R code to fully reproduce our results are available in a public repository: 10.6084/m9.figshare.23691579.

## RESULTS

### Comparing various taxonomic assignment and database building strategies

Figure 1 shows an overview of the performance at the species level (accuracy) of the 48 different "strategies" tested for the two barcodes (*rbcL* and ITS2) and for local *vs.* foreign species (*i.e.* a total of 192 combinations). A similar graph for each taxonomic level is available in the Supplements (section 3.1). In the remainder of this section, these results are broken down into smaller and more digestible pieces by focusing on one aspect of these "strategic choices" at the time.

#### Comparing four blast taxonomic assignment approaches

The methods using the most frequent taxon among the top 10 blast hits (TopN, TopNPlus) never outperformed the methods using the best blast hit (TopHit, TopHitPlus) and, in many situations, they performed worse (Figs. 1 and 2).

The method we called TopHitPlus outperformed the simpler TopHit approach in several circumstances, particularly for *rbcL* at species level (up to 13.6% more correct identifications at species level when restricted sequences from local species were blasted against a world general database). The maximum benefit for ITS2 reached 3.9% better prediction for TopHitPlus relative to TopHit. The TopHit approach never outperformed TopHitPlus or with a very limited improvement (maximum 0.3% of better assignments). The *rbcL* barcode is less specific and ties between the highest bit scores are more frequent than for ITS2. In this case, preferring the most frequent taxon (as performed by TopHitPlus) provides a slight advantage.

For the remainder of this article, we will concentrate on the results of this TopHitPlus approach.

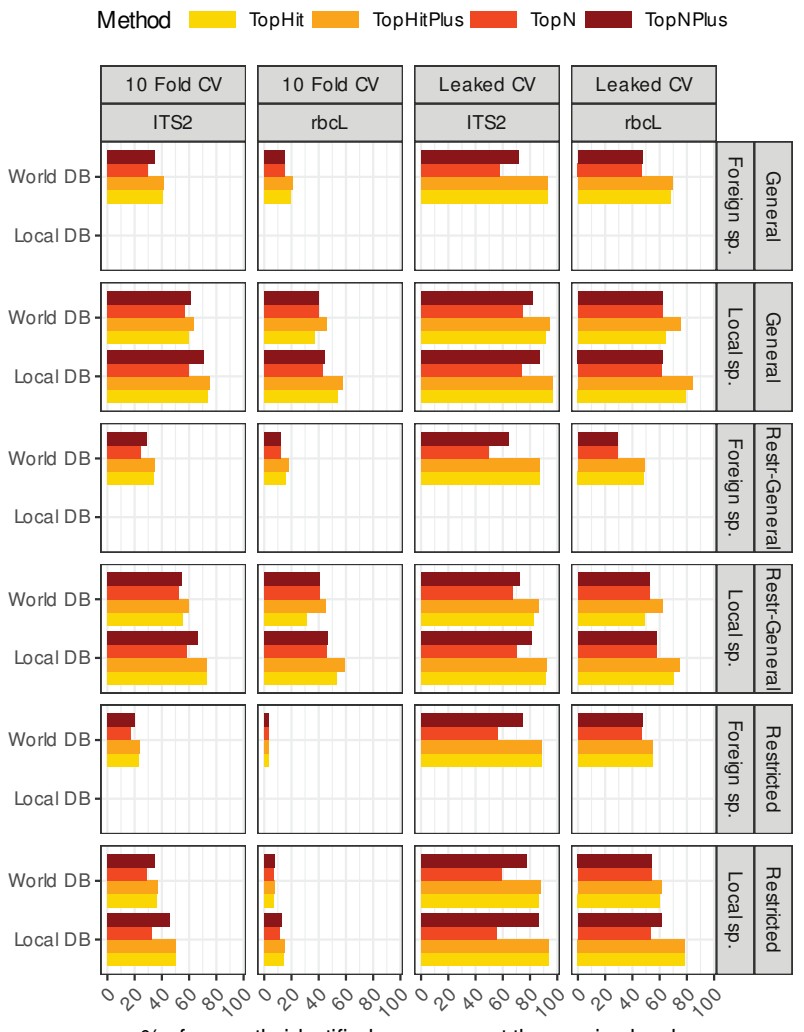

**Figure 1** **This rather dense figure is broken down into more digestible chunks in Figs. 2–4 (which also include results at higher taxonomic levels).** The primary objective here is to provide a one-figure summary of all combinations of strategies evaluated in this study: we assessed two barcode sequences (ITS2, rbcL) using two cross-validation strategies, where the blasted sequence is still present in the reference database (Leaked CV) or not (10 fold CV) and with four ways to assign the taxonomy with blast (four colours). We examined the effect of blasting a local species or foreign species sequence against either a database containing only the local flora (Local DB) or all available sequences (World DB). We also explored outcomes when blasting (1) full-length sequences against a database of full-length sequences (Global), (2) sequences restricted to a specific primer-amplified region against a database of restricted sequences (Restricted), or (3) restricted sequences against a database of full-length sequences (Restr-General).

### Leaked CV vs. 10-fold CV

The accuracy estimates obtained with leaked cross validation (CV) were better than with the 10-fold CV approach (Fig. 3). However, even though in that case, the true sequence is present in the reference database, blast had still difficulties retrieving the correct species level identification among all the other sequences in 4–14% of the cases for ITS2 and in 16–53% of the cases for *rbcL* (depending on the strategy used). The lower accuracy for *rbcL*

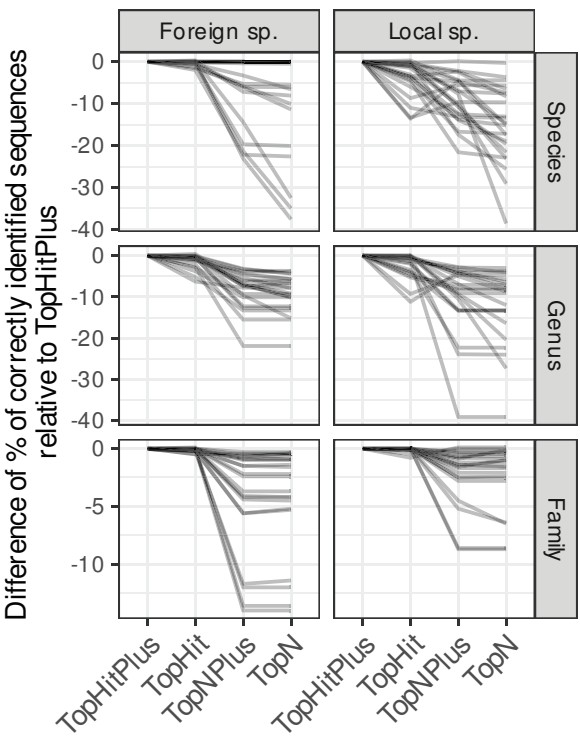

**Figure 2  Comparison of the relative performance of four assignment strategies using blast.** Each grey line represents the difference in accuracy obtained between the four assignment methods (with 'TopHitPlus' as baseline reference) for one of 24 combinations of cross-validation strategies (*e.g.*,: 10-fold CV, general DB, world DB and ITS2). Details at the species level are presented in Fig. 1. 'TopHitPlus' almost always performed as well or better than the other methods.

is most likely due to the lower specificity of this barcode which makes the distinction between closely related species more difficult.

As expected, 10-fold CV (left panels on Fig. 3) accuracy values were clearly worse than for leaked CV. In real biological samples, we can expect that the exact target sequence will sometimes be present in the reference database and sometimes not. So the accuracy in real biological samples is probably intermediate between the results obtained with 10-fold CV and leaked CV. In an unpublished dataset based on 354 real pollen samples, we observed a 100% identity match for only 40.3% of 12,971 ITS2 ASVs and 64% of 13,132 rbcL ASVs. So in practice the exact sequence does not seem to be always present in the reference database (or at least not matched). In the remainder of this article, we will therefore concentrate on the 10-fold CV results which represent a more realistic scenario.

### Restricted vs. general databases

Using 10-fold CV, the accuracy obtained with the database restricted to the region amplified by a specific primer was systematically worse than with the general database containing the full-length, untrimmed sequences (see left panels on Fig. 3, yellow lines). This is probably because these restricted databases contain much fewer sequences after
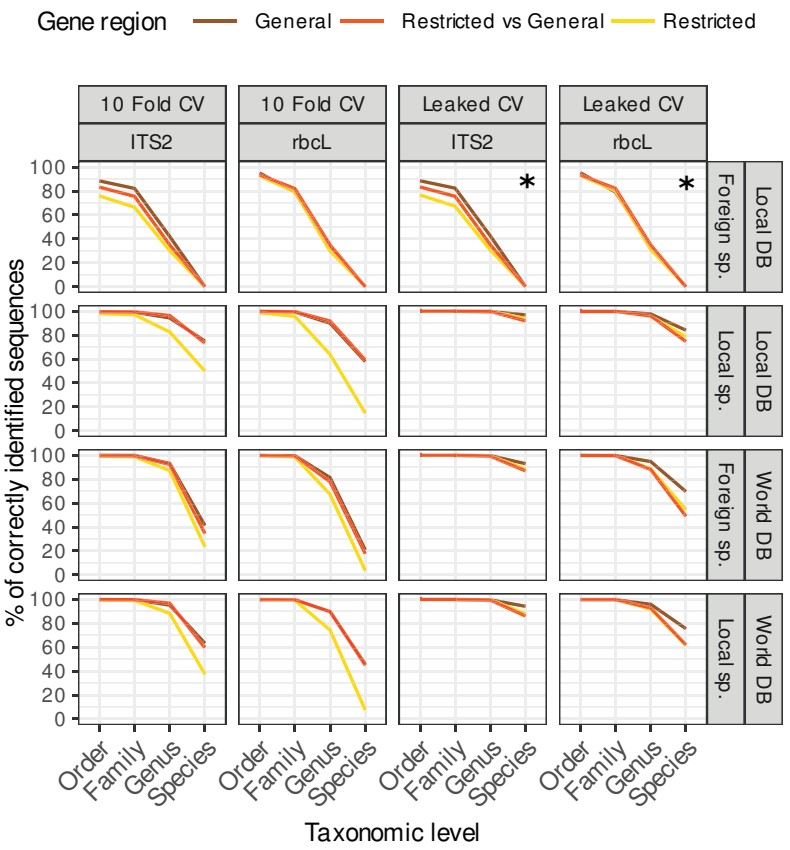

**Figure 3 Accuracy for 'TopHitPlus' results only.** The purpose of this figure is mainly to compare the results obtained with leaked CV (right) *vs.* 10-fold CV (left) on the one hand, and the restricted, general and restricted-general databases (colored lines) on the other hand. Note that the two top right facets marked with a star (*) represent a particular case where foreign species sequences were blasted against a local database with a leaked CV design. In this setting, the target foreign sequences have been removed from the reference database, so these results do not really represent a true leaked CV approach and are in fact quite similar to the corresponding top left panels showing the 10-fold CV results.

trimming, cleaning and dereplication. So, once the true sequence is removed from the reference database, blast struggles to find alternative sequences taxonomically close to the true target sequence. Other studies on different barcode genes have also reported a decrease in taxonomic accuracy when using reference databases with sequences trimmed to a specific primer pair, *i.e.* on COI gene for Arthropoda and Chordata (*Robeson et al., 2021*) or on fungal ITS gene (*QIIME2 Documentation, 2022*).

When restricted sequences were blasted against a general database, the accuracy was similar to the one obtained when general databases were cross-validated against themselves (Fig. 3, left panels, red and brown lines). We expected that the longer sequences in the general *vs.* general cross-validation would artificially increase the accuracy thanks to the additional DNA portions providing more information to distinguish closely related species. However, this is not what was observed from the data. When we use the general database, blast probably struggles to find alternative sequences (once the true sequence has been removed by the cross-validation process) for many very long sequences.

 

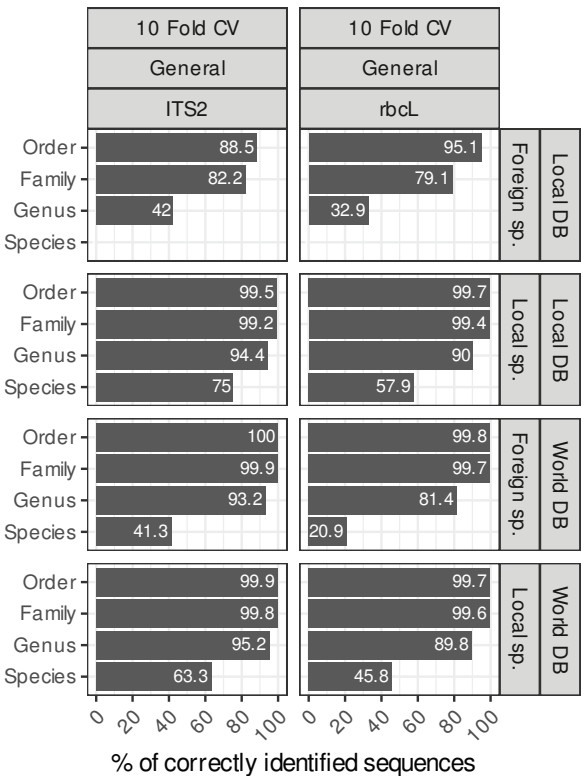

**Figure 4 Accuracy for TopHitPlus, 10-fold CV and general database only.** The main purpose of this figure is to compare the results between a local database (DB) and a world DB for both local and foreign species at various taxonomic levels.               

In the remainder of this article, we will therefore focus on the results for the general database CV as a more general case with more tested sequences.

### Local and foreign species vs. local or world databases

Blasting the sequences of local species (*i.e.* species present in Belgium in our study case) against a local reference database (*i.e.* containing only sequences of Belgian species) instead of a world database (*i.e.* with all NCBI sequences) increased the species-level accuracies by ∼12% for both *rbcL* and ITS2 (Fig. 4). For ITS2, the accuracy rose from ∼63% to ∼75% when we used a world database or a local database respectively. For *rbcL*, the accuracy rose from ∼46% to ∼58%. Despite this improvement, the error rate at species-level is still rather high (25% for ITS2 and 42% for *rbcL*). Of course, when sequences of foreign species were blasted against a local database, the accuracy at the species level was 0%, while when using a world database, the accuracy increases weakly to 41% for ITS2 and 20% for *rbcL*. So, the identification of foreign species at species level was always poor even when a world database was used. This is probably due to the fact that this foreign flora is not as well represented in the databases as the local flora considered in this study case (*i.e.* Northern Europe flora). So once the true sequence was removed from the database, blast struggled to match alternative sequences that are close to the real sequence.

However, at higher taxonomic levels, using a world database is the best option in all cases. Indeed, for local species, using a local or world database made little difference at the

genus level (∼94% accuracy for ITS2 and ∼ 90% for *rbcL*; Fig. 4). For foreign species, however, using a world database caused a substantial improvement (∼93% accuracy for ITS2 and ∼81% for *rbcL*). If a local database is used instead, these accuracies for foreign species dropped to ∼41% for ITS and ∼32% for *rbcL*. At the family level, the accuracy was ∼99% when a world database was used, for both foreign and local species.

## Establish rules for deciding the reliability of identifications based on identity and consensus scores

### Generalized linear models (GLMs)

As expected, the probability of correct identification increased when both the identity and consensus scores increased at all taxonomic levels (Fig. 5). Local and foreign species followed a similar trend with, on average, a higher probability of correct identification for the local species. However, there was no perfect separation line between correct and incorrect identification based on the combination of these two scores. Even with a 100% identity score, many of the identifications remained wrong at the species level. Even when the consensus score is as low as 10%, 20–40% of the identifications remained correct at species level. This means that as soon as we set a consensus or identity threshold under which we do not trust the identification, we will also discard a lot of "correct" identifications.

### Classification trees

Classification trees can help define simple rules based on the identity and consensus scores. Instead of a GLM, we fit a classification tree, with the correct/incorrect identification as response and the consensus and identity scores as predictors at each taxonomic level, for each barcode and World *vs*. Local databases. An example is displayed on Fig. 6 for *rbcL* at species level (see Supplements section 4.3 for the trees corresponding to other cases). The nature of this recursive partitioning algorithm makes it particularly well suited to "detect" best thresholds (rather than linear increases). A visual examination of figures like Fig. 5 with univariate binomial GLMs is also helpful in making better-informed decisions on the best thresholds.

The rules "proposed" by the tree must always be adapted depending on the type of errors one favours. For example, here, we have established a set of rules quite stringent at the species level. The aim was to eliminate as many false positives as possible since we know that the percentage of correct identification is rather low at the species level. On the contrary, much less stringent rules were chosen at the genus level. The error percentage would already be minimal if we accepted all the identifications at the genus level (particularly for ITS2). So, the aim was to eliminate only the most obviously doubtful identifications.

Based on the results from the classification trees (Fig. 6) and the univariate GLM graphs (Fig. 5), the following rules were decided: at the genus level, we used only the World DB results and discarded the taxonomic assignments with Consensus < 40% and Identity < 98.5% for ITS2 or Identity < 99.7% for *rbcL*. At the species level, for ITS2, with a Local DB, we discarded all predictions with Identity < 99%. Then, for the sequences whose

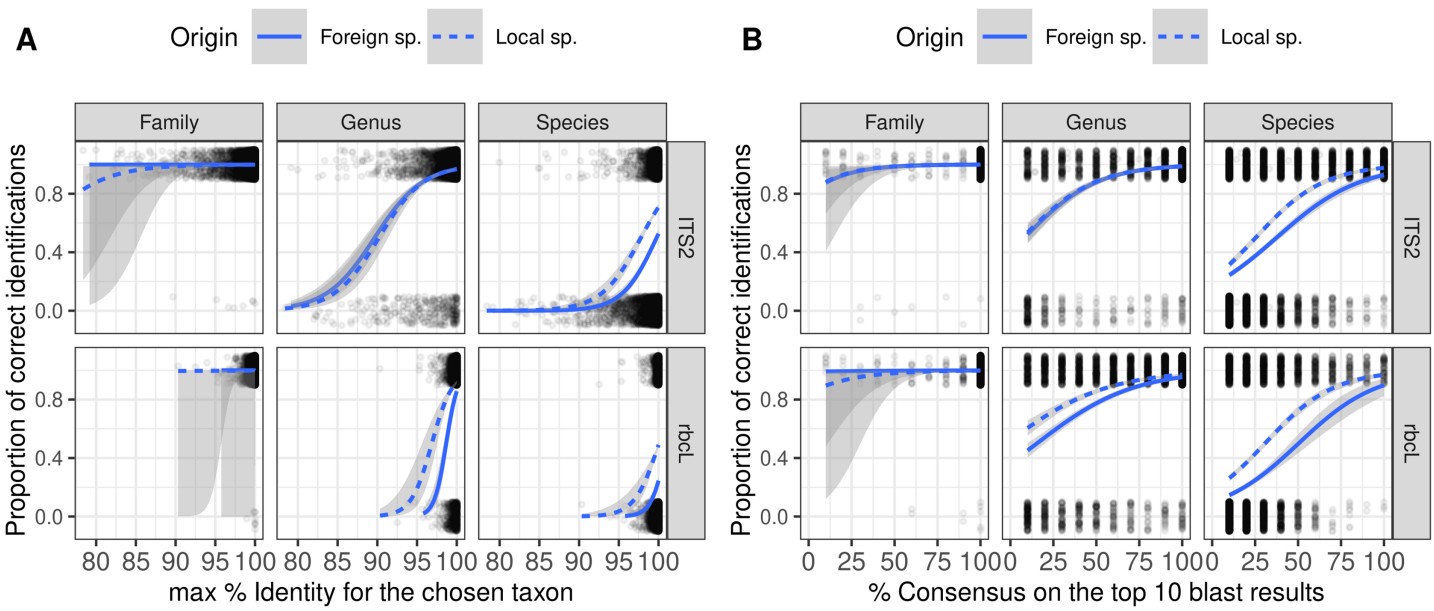

**Figure 5 Figure showing the relationship (Binomial GLM) between the proportion of correct identification and the identity score (A) or consensus score (B) for three taxonomic levels and two barcodes.** The probability of correct identification increases when the identity score and the consensus score increase, in a similar way for both local (dashed blue lines) and foreign species (plain blue lines). But even with 100% identity, many identifications are incorrect at the species level. We show here the results for TopHitPlus, on a world, general database and 10-fold CV. See Supplements, section 4.1.2, for similar graphs on a local database (similar conclusions).

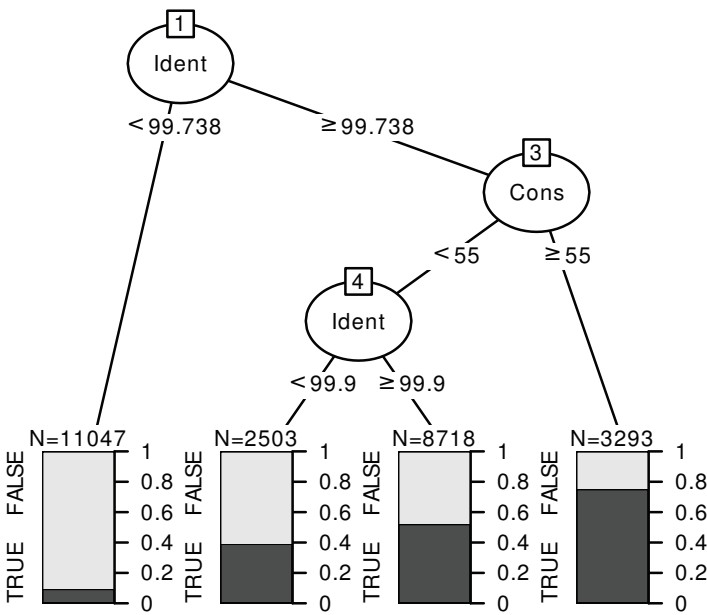

**Figure 6 Example of a classification tree for rbcL at species level.** The response is a binary variable indicating whether the identification is correct (TRUE) or not (FALSE), without distinction between foreign and local species, and with two predictors: the identity score ('Ident') and consensus score ('Cons') at species level. We used the data for TopHitPlus, on a local, general database and 10-fold CV. The trees for ITS2, world databases and at the genus level are provided in the Supplements section 4.3.

identification was discarded with the local database approach, we considered the results from the World DB and discarded all sequences with a consensus < 40% and an Identity < 99.9% or a consensus < 20% (even if the identity was 100%). For *rbcL*, with a Local DB, we discarded all predictions with Identity < 99.7% or with an Identity <= 100% and a consensus < 50%. Then, we also used the World DB and discarded all sequences with a consensus < 30% or a consensus < 60% and an identity < 100% (NB: such an approach combining Local and Global databases was for example used by *Bell et al. (2020)* or *Casanelles-Abella et al. (2022)*).

### Reduction of prediction errors after applying the rules based on identity and consensus scores

Figure 7 shows what happened when these rules were applied in our study case. The percentage of sequences wrongly identified at the species level dropped from 37–80% to 5–17%. However, this comes at a cost: the sequences without identification rose from 0% to 17–92%. For foreign species with *rbcL* for example (worst case), the percentage of prediction error was 80% (on a World database) before filtering out low quality assignments. After applying these rules, the percentage of errors dropped to 5.5% but the percentage of unassigned sequences skyrocketed at 91.5%, with only 2.9% of the sequences correctly identified. Even for the best case (local species with ITS2), after filtering, ∼17% of sequences were unassigned, ∼68% were correctly identified, and ∼17% displayed wrong identifications. This means that, even after filtering, 20% of the assignments (17/(17 + 68)) were still incorrect at the species level. At the genus level, the gains were marginal (the raw predictions were already good enough). The most important change concerned the assignment of foreign species with *rbcL:* the error rate dropped from ∼19% without rules to 12% after applying the rules. In conclusion, in our study case, while the identification accuracy at the genus level was satisfactory, it seemed difficult to remove the wrong identifications at species level without also removing most of the correct ones, even by carefully choosing the consensus and identity thresholds.

## Comparing *rbcL* and ITS2 for various botanical families

The literature regularly recommends using a combination of several barcodes. ITS2 and *rbcL* are often used and cited as good choices for plant metabarcoding (*Bell et al., 2022*). However, the results presented so far show that *rbcL* systematically performed worse than ITS2. However, these conclusions are based on global estimates over the whole taxonomic spectrum and the *rbcL* barcode might still perform better than ITS2 for certain taxa.

Figure 8 compares the percentage of correctly predicted sequences at the species and genus level for a selection of plant families frequently represented in the target sequences or important as food resources for bees. We observed a better performance of *rbcL* at the species level only for the Balsaminaceae and Pinaceae. In all other families and at the genus level, ITS2 performed almost as well or better than *rbcL*. The percentage of correctly predicted sequences varied a lot between families, particularly at species level. The lowest performance was observed for Salicaceae, Grossulariaceae, Pinaceae, Balsaminaceae, Rosaceae, Geraniaceae (35–61% of correctly predicted sequences for ITS2 but with better

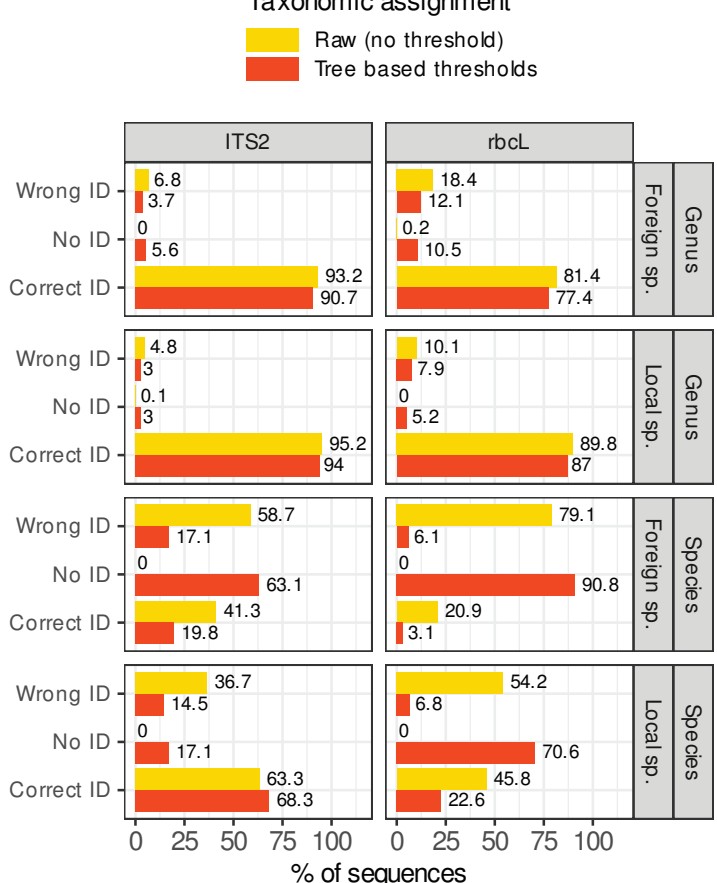

**Figure 7** Percentage of sequences for which the identification was correct, wrong or for which no taxon was assigned (No ID) before (in yellow, assignment = raw) or after (in red, assignment = "Tree-based thresholds") filtering out the taxonomic assignments that were considered as not trustworthy based on a series of rules inspired by the classification trees results and GLMs graphs. Ideally, all yellow wrong IDs should become red "No ID" and the yellow and red correct IDs should remain as close to each other as possible.

results with *rbcL* for Balsaminaceae and Pinaceae) while the best performances were observed for Sapindaceae, Rhamnaceae, Hydrophyllaceae, Lythraceae, Ericaceae and Hypericaceae (95–85% of correct predictions). The accuracy was less variable at genus level. The families performing the worst at the genus level with the ITS2 barcode are Malvaceae, Apiaceae and Borraginaceae, with 85–90% of correct predictions. Note that in Fig. 8, we focused on Local species and that the number of sequences targeted can be rather low for certain families: 9–16 ITS2 sequences and 7–30 *rbcL* sequences for families Hydrophyllaceae, Grossulariaceae, Balsaminaceae, Rhamnaceae and Lythraceae.

# DISCUSSION

Using cross-validation on taxonomic reference databases has proved very instructive in several respects. This approach can help to decide on the best strategies for taxonomic assignment and database building, but also to estimate the reliability of the taxonomic assignments and define thresholds below which these taxonomic assignments should not

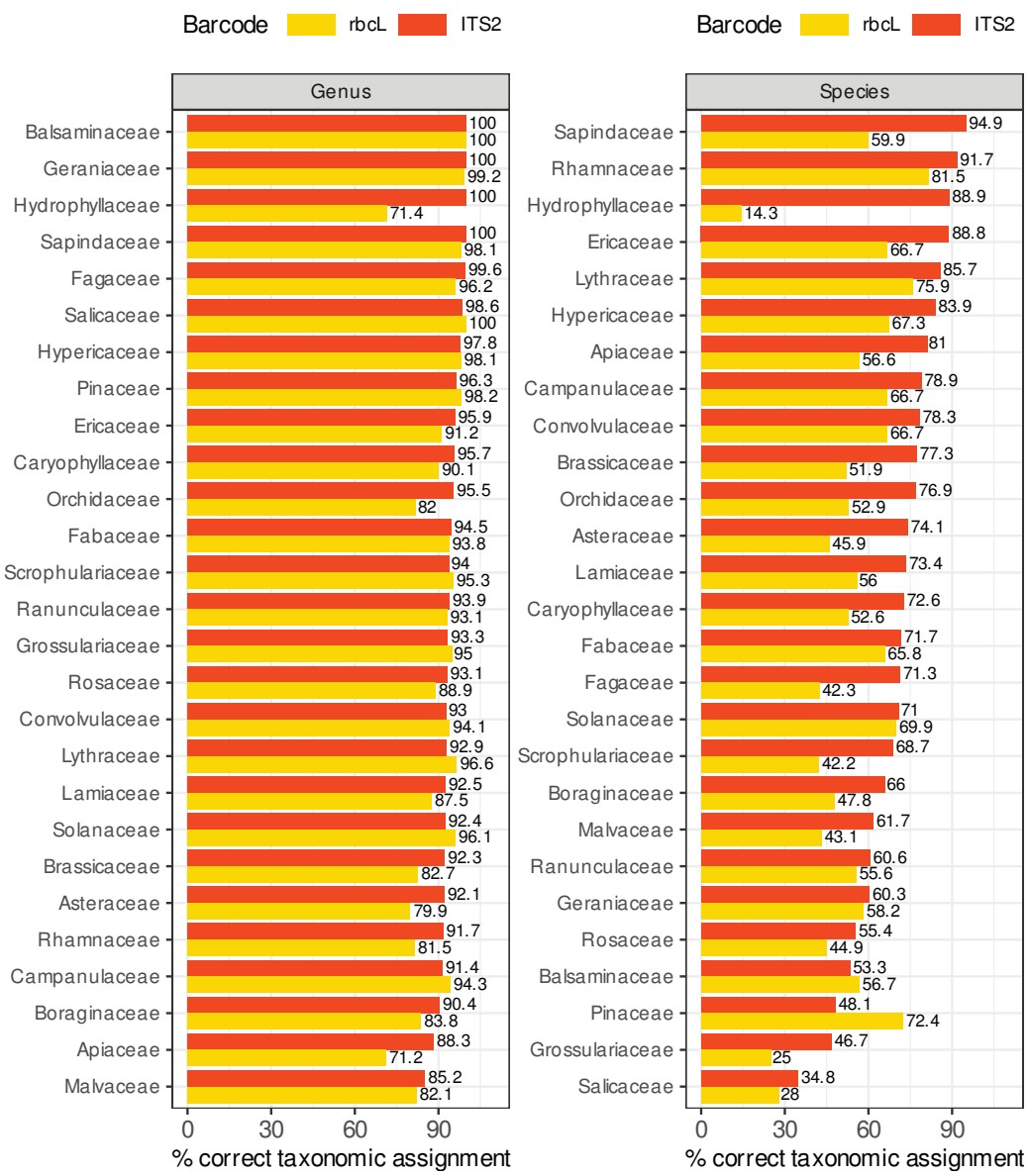

**Figure 8 Comparison of ITS2 and rbcL barcode performances at genus and species level for a selection of families well represented in reference databases and/or important as food resources for bees.** The results displayed concern 10-fold CV, general DB, TopHitPlus, local DB and local species. Supplements, section 5.1 show that the results for a world DB including or not the foreign species are quite similar.

be trusted. Cross-validation can be done *in silico*, at no extra cost, even before samples are collected, using the reference databases that are needed for the actual metabarcoding process.

Concerning the best taxonomic assignment strategy, our results showed that the methods based on the best bit score (itself based on the Identity score) always performed the best in our study case, compared to approaches based on the best consensus. However, computing the Consensus score can be helpful to break ties among the best blast hits (*i.e.* with identical bit scores) but also to estimate the reliability of the taxonomic assignment.

We observed that when the true sequence is present in the reference database (leaked CV), BLAST struggled to find the right match in ~10% (ITS2) to ~20% (*rbcL*) of the cases because of the presence of similar sequences in the databases belonging to other taxa.

Blasting against itself a reference database restricted to the amplified region of the barcode genes caused drops in taxonomic prediction accuracy, probably because the number of sequences was dramatically lower in these restricted databases. It is better to keep a more extensive reference database with full-length sequences. One could expect that blasting shorter sequences (*i.e.* the ones clipped to match the real amplicon of our study case) against a general database with the complete sequences would decrease the accuracy of the predictions. However, we observed only a minimal accuracy drop at the species level in that case. This is probably because the targeted amplicons are already among the most polymorphic and discriminant portions of the chosen barcodes. This also means that increasing the amplicons size while remaining in the same DNA region (*i.e.* ITS2 and *rbcL*) would probably not greatly improve the taxonomic resolution. However, new techniques like MinION nanopore sequencing might allow the amplification of much larger sequences in more polymorphic DNA regions and could, therefore, allow better taxonomic resolution if we can reconstruct the corresponding reference sequences libraries.

Globally, the taxonomic assignments were excellent up to the genus level, particularly for ITS2 (5–7% of wrong taxonomic predictions) and for both local and foreign species. Using a Local reference database makes little sense at these higher taxonomic levels because it does not increase the prediction accuracy for local species but strongly decreases the prediction accuracy for foreign species. At the species level, the chosen strategy had a higher impact on the accuracy of taxonomic assignment: using a Local database improved the prediction quality for local species to reach a minimum but still rather high level of 25% and 32% of wrong predictions for ITS2 and *rbcL* respectively. The results were even worse when foreign species sequences were blasted even against a world database (59% and 80% of wrong taxonomic assignments for ITS2 and *rbcL*, respectively). This shows the importance of the reference database, which must contain as many reference sequences as possible.

These relatively poor results at the species level also stress the importance of defining rules to decide whether a taxonomic assignment should be considered. Our results show that the percentage of correct identification at the species level was still rather poor, even when the identity score was close to 100%. In contrast, when the consensus score was high (>70%), identifications at the species level were almost always correct. However, a low consensus score does not necessarily mean that the identification is incorrect. The best threshold values will depend on the type of errors (false positive, false negative) that are the most problematic for the study or even for each scientific question. Applying a set of rules to discard dubious taxonomic assignments caused a considerable reduction of prediction errors but at the cost of a high proportion of unassigned sequences at the species level. Among the assigned sequences, the ratio between correct and incorrect identification was still not very good (4 to 1 in our best case). The only way to improve these scores would be to increase the taxonomic coverage and diversity of sequences in the databases for all species worldwide or to target more polymorphic DNA regions.

We also described the variability of the prediction accuracy between plant families. *rbcL* rarely outperformed ITS2, but using both barcodes could still be helpful to overcome problems to which cross-validation is blind, such as poor taxonomic coverage of the databases or amplification biases. In addition, interesting complementary between both barcodes were noticed, like for the Balsaminaceae and especially the Pinaceae families, for which ITS2 performed poorly, whereas *rbcL* showed much better assignment accuracies.

Some of our methodological choices could have biased the accuracy estimates. For example, we discarded the hybrids from the target sequences evaluated, mainly for technical reasons. The taxonomic nomenclature of hybrids is even more complex than for regular species and synonyms between or even within databases would be challenging to manage. The presence of hybrids in real-life biological samples could decrease the accuracy at least at the species level.

Using the estimates from 10-fold CV is also disadvantageous for species poorly represented in the database or with little intraspecific variation. The dereplication step during database building will turn these multiple identical sequences into a unique sequence in the final reference database. In the worst case, if a species is represented only by one sequence in the database, blast will never be able to match the correct species with 10-fold CV. In real-life conditions, the accuracy is probably intermediate between the 10-fold CV and the leaked CV estimates because sometimes the exact sequence will be present in the reference database and sometimes not.

Cross-validation also has at least two major blind spots inherent to this approach that could lead to overestimating the capacity to correctly identify the species composition from biological samples. (1) Amplification biases. In biological samples, the DNA of certain taxa might not be amplified enough during the PCR for example due to competition with other species sequences or if there are mutations in the primer regions. This could induce a lot of false absences that are not simulated by the *in silico* cross-validation process. Using multiple or degenerate primers could decrease these amplification problems. (2) Completeness of the reference databases. The accuracy estimated with cross-validation only considers sequences for taxa that are present in the reference database. If a large portion of the real-life target species is absent from the original database, cross-validation will overestimate the accuracy that can be expected when carrying out the taxonomic assignment of metabarcoding sequencing reads. In our study case, the taxonomic coverage is relatively good (∼80% of the species and ∼95% of the genus present in our study area) but this is not always the case. For example, *Braukmann et al. (2017)* estimated *via* cross-validation that the % of correctly identified sequences was ∼80% for matK, ∼72% for ITS2 and ∼45% for *rbcL*. However, their matK and ITS2 databases covered only ∼60% of the flora present in the 28 Canadian national parks from their study (∼95% for *rbcL*), so the true accuracy for real biological samples will probably be lower for matK and ITS2 barcodes because the species missing from the databases will not be correctly predicted (at least at the species level). Blasting foreign species on a local database can give an idea of what happens in this case. We have seen that even at the genus level, the drop in accuracy can be severe in this rather extreme case. However, this problem should become less important as the reference databases grow.

Cross-validation results can also highly depend on how the reference database was built. For example, there are various options available to dereplicate the sequences. When the same sequence is attributed to different taxa in the non-dereplicated database, one could choose a majority mode and label each identical sequence with the most frequent taxon. This could hide a good part of the taxonomic assignment uncertainty and would undoubtedly lead to overestimating the accuracy, at least at the species level. In this study, the databases were built in "unique" mode, where identical sequences displaying different taxonomies are all conserved with their respective taxonomic labels (*Dubois et al., 2022*).

Using different barcode markers on the same samples could help work around problems related to both amplification biases and the completeness of the reference database, as the weaknesses of one barcode could be compensated by the other. So, even though *rbcL* performed almost systematically worse than ITS2 in our study case, using both barcodes might still be interesting.

Thus, various factors can cause underestimation (*e.g.* 10-fold CV) or overestimation (*e.g.* hybrids, low taxonomic coverage, amplification problems) of taxonomic assignment quality estimates and the values obtained should be seen as indicative.

## CONCLUSION

Our study provides a comprehensive example of how cross-validation can be used as a simple yet powerful tool to make better-informed decisions about many aspects of the metabarcoding process:

1) Assessing the relative accuracy of identifications: for example, in our study case, ITS2 systematically outperformed *rbcL*, and errors in genus-level identifications were consistently low. However, the risk of incorrect identification at the species level remained substantial, even with identity scores as high as 100%. The possibility to test many taxa also revealed that species-level identification may be more reliable for certain plant families than others.

2) Selection of thresholds to distinguish reliable from dubious identifications: modelling the % or errors *vs* identity and consensus scores provides invaluable help in making more rational choices about these thresholds. Despite stringent filtering to eliminate uncertain assignments, we found that at least one in five sequences deemed suitable for species-level identification ultimately proved to be identified incorrectly.

3) Arbitrating between different strategies (*e.g.* database construction, taxonomic assignment, *etc.*): we have shown, for example, that limiting the reference database to the local flora had a positive but rather limited effect on accuracy for local species, while severely reducing accuracy for exotic species.

We have also acknowledged the limitations of this approach: inability to account for lack of taxonomic coverage (species missing from the database), blindness to amplification biases commonly encountered in metabarcoding, and sensitivity to the way the database is constructed and evaluated (*e.g.* dereplication, trimming of sequences, cross-validation procedure). Consequently, despite its lower ability to discriminate closely related species,

*rbcL* may still be a useful barcode as a complement to ITS2, for example because of a better taxonomic coverage of its databases or different amplification bias patterns.

Furthermore, the results presented here only apply to our study case. Different taxonomic groups, geographical regions and barcode sequences may lead to different conclusions. Because of its simplicity and affordability, cross-validation appears to be a method of choice to revisit the optimal strategy in each study case, while wet-lab validation approaches (mock communities, real biological samples) remain very complementary, for example for the characterization of amplification issues, the evaluation of detectability and quantification, *etc*.

### Funding
This work was part of the PolBEES project funded by the Walloon Agricultural Research Centre (Moerman act). The funders had no role in study design, data collection and analysis, decision to publish, or preparation of the manuscript.

### Grant Disclosures
The following grant information was disclosed by the authors:
PolBEES Project funded by the Walloon Agricultural Research Centre (Moerman act).

### Competing Interests
The authors declare that they have no competing interests.

### Author Contributions
- Gilles San Martin conceived and designed the experiments, performed the experiments, analyzed the data, prepared figures and/or tables, authored or reviewed drafts of the article, and approved the final draft.
- Louis Hautier conceived and designed the experiments, performed the experiments, authored or reviewed drafts of the article, and approved the final draft.
- Dominique Mingeot conceived and designed the experiments, performed the experiments, authored or reviewed drafts of the article, and approved the final draft.
- Benjamin Dubois conceived and designed the experiments, performed the experiments, analyzed the data, authored or reviewed drafts of the article, and approved the final draft.

### Data Availability
The data and R code for detailed analysis are available at figshare: San Martin, Gilles; Hautier, Louis; Mingeot, dominique; Dubois, Benjamin (2023). How reliable is metabarcoding for pollen identification? An evaluation of different taxonomic assignment strategies by cross-validation. Figshare Dataset. https://doi.org/10.6084/m9.figshare.23691579.v1.

A part of the code has also been encapsulated in an R package for better documentation and reusability. This is available at GitHub and Zenodo:

- https://github.com/GillesSanMartin/CVrefDB.
- Gilles San Martin. (2023). GillesSanMartin/CVrefDB: PeerJ Publication Version (v1.0.0). Zenodo. https://doi.org/10.5281/zenodo.10121058.

## Supplemental Information

Supplemental information for this article can be found online at http://dx.doi.org/10.7717/peerj.16567#supplemental-information.

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
