# Peer review of "How reliable is metabarcoding for pollen identification? An evaluation of different taxonomic assignment strategies by cross-validation"

_PeerJ, doi:10.7717/peerj.16567_

## Round 0.1 · original submission · Minor Revisions

Dear authors,

Please address the reviewers' concerns. More concise presentation and clearer explanations are called for.

·

Basic reporting

While the experimental design is sound and the study conducted will be beneficial to the field, this manuscript could use additional editing to remove overly redundant sections. The writing is currently quite verbose, making it difficult for the reader to comprehend. For example, many of the issues described in the introduction have already been discussed in Dubois et al 2022. What can be removed here to decrease introduction length? The paper could be reduced to just a few pages discussing potential issues that arise with analysis choices, generally describe the types of tests the authors did, and what is the best approach and why.

Examples in the methods section include:

Line 147: "We downloaded all sequences available on the National Center for Biotechnology Information (NCBI) and cleaned it up through the DB4Q2 pipeline detailed in Dubois et al. (2022)." could instead read "All available sequences were retrieved from the National Center for Biotechnology Information (NCBI) and cleaned with the DB4Q2 pipeline (Dubois et al., 2022)."

Line 151: the part about "and a last optional step consisted in the database restriction" could be removed, since it is redundant given the proceeding information in this sentence.

Line 273: "this represents only < 2 % of the sequences and only for some particular cases" could instead read "representing <2% of the sequences for particular cases"

There are also sentences which could be rearranged, or words that could be exchanged for others, to improve reader comprehension.

Examples in the discussion:

Line 502: "The situation was more contrasted at the species level" - what does this mean?

Line 514: "In the end, the scientist must choose..." this seems like a sentence to close out on, not to place in the middle of the discussion.

Line 587: "We used a slightly unusual approach with blast." This sentence doesn't add any useful information.

There are many other examples of using words that don't add information to the sentence, like usage of "very", "more and more", "less and less", etc.

Finally, overall, the conclusion is too long. Some of this information could be moved to the discussion. The conclusion should instead highlight the main points of the paper in 4-5 sentences and offer a sentence or two to connect the paper with the overall body of literature.

Experimental design

The experiments are well designed and very thorough. The authors have tested every combination of variation in analysis practices for metabarcoding data. Those who do this type of research will find this study incredibly useful, and the authors comment on the gaps in the literature concerning best practices for metabarcoding data analyses. The methodology is described sufficiently to allow others to reproduce the same results.

Validity of the findings

There are differing opinions in the field about how best to minimize bias or false positives/negatives when conducting this research, so this study will benefit the body of literature concerning metabarcoding data analyses practices. The supplementary material is particularly useful for those looking to follow along with this study as a guide to improve their own data analyses.

Additional comments

Minor comments:

Line 17 and elsewhere: is this meant to be “parameter values”?

Line 35 and elsewhere: there are unnecessary spaces between punctuation marks and the preceding words, e.g., “rather high : 25%” should be “rather high: 25%”; Lines 127-140 many “?” have unnecessary spaces; etc

Line 50: “plants foraged on *and* the nutritional quality”?

Line 51: what are “contamination pathways”?

Line 51 - Why the ellipses?

Line 54 - writing seems a bit informal here with the “etc.”

Lines 54-58: Begin the second paragraph with the sentence beginning in line 54.

Line 59: “different sequences for barcoding can be used.” What is a “barcode”? You do not define this term.

Lines 85-86 - How does Blast+ perform better than other algorithms?

Lines 92-94 - How? Please elaborate.

Line 96: “one *needs* to test”

Lines 127-140 - This paragraph would be easier to interpret if research questions were grouped into overarching questions (3 - 4) and then elaborated on in each section.

Lines 154, 277, and elsewhere: it is unnecessary to have 1 sentence paragraphs. This can be incorporated into the preceding paragraph.

Lines 165 - 167 - Why would having longer sequences with which to distinguish taxonomic predictions be considered a bad thing?

Sections 2.1.3 and 2.1.4 - What is the difference between Foreign vs. local and local db vs. world db?

Figure 1. If this graph is split later in the paper, this figure may be unnecessary or could be moved to the supplemental materials. It is currently very complex and hard to draw any conclusions by comparing each experimental combination.

Line 339: why focus on a worse-case scenario?

Lines 483-486: in cases where assignments cannot be made due to sequence similarity of barcodes for different species, couldn’t you just increase the bit score threshold to break ties (assuming there is one with a higher bit score?)

Line 465: “best strategies *for* taxonomic assignment”



Other questions that could use clarification at other points in the manuscript:

What happens if you increase or decrease the database size? i.e., does this lower or raise the likelihood of false positives or negatives?

If sequences aren't trimmed to the amplification region is there a higher rate of false positives?


This paper was reviewed by Makaylee Crone with Sean Bresnahan.

Reviewer 2 ·

Basic reporting

Clear, unambiguous, professional English language used throughout.
Yes, generally well-written. There are few typographical errors to correct (some sentences ended with “…”)

Intro & background to show context.
The introduction is well written but I found some of the detail irrelevant to the hypothesis being tested. This is an exploration of the performance of BLAST on a couple of gene regions against different database scenarios, and microscopy is not relevant here.

Literature well referenced & relevant.
Yes it is well referenced, although some sections do not seem very relevant.

Structure conforms to PeerJ standards, discipline norm, or improved for clarity.
I will leave this to the editors.

Figures are relevant, high quality, well labelled & described.
Yes, figures are high quality and relevant. However, there were so many scenarios being tested that some figures required a better written summary to quickly understand them.

Raw data supplied (see PeerJ policy).
Yes, raw data was supplied.

Experimental design

Original primary research within Scope of the journal.
I will leave this to the editors to decide.

Research question well defined, relevant & meaningful. It is stated how the research fills an identified knowledge gap.
I think the research questions could be more clearly defined. At its core, this study evaluated whether BLAST performs better when a query sequence is present in a database, and tests two genetic markers and a number of different restricted database scenarios. I think the manuscript could be greatly simplified by removing much of the detail about pollen metabarcoding and just focusing on the performance of BLAST against different query parameters and the database construction. The research questions could then be more clearly and simply stated around this aim, with a brief discussion about how these scenarios are applicable to pollen metabarcoding.

Rigorous investigation performed to a high technical & ethical standard.
Yes, the analyses were performed to a very high technical standard.

Methods described with sufficient detail & information to replicate.
Most of the methods had a high level of detail and all the information required to replicate. I commend the authors for their care and transparency in providing their methods and results. One area that lacks detail and may confuse the reader is how the data was generated for the cross-validation. The manuscript is based on an in-silico simulation of bee foraging. I think it needs to be made more obvious at the start that the data are sequences randomly taken out of databases.

Validity of the findings

Impact and novelty not assessed. Meaningful replication encouraged where rationale & benefit to literature is clearly stated.
Not quite clear what is required from the reviewer in this section.

All underlying data have been provided; they are robust, statistically sound, & controlled.
Yes, I was able to download all the supporting data.

Conclusions are well stated, linked to original research question & limited to supporting results.
The authors did not have strong conclusions other than to say every situation is different. This section needs more clarity.

Additional comments

Introduction
Line 51: remove … or finish sentence
Line 55: consistent author naming?

Methods
Can you explain why the clipped databases were so much shorter than the full-length databases? Was the relevant ITS2 and rbcL region part of the full-length database? If some sequences did not overlap with the target region, then it is hard to compare the results between a database containing only the target region vs a longer version that encompasses the target region.

I am not sure mock communities vs. cross validation is directly comparable. The purpose of mock communities is often to test detectability and quantification of different species composition. Microscopy does relate to pollen identification, but reliability of taxonomic assignment depends highly on human expertise. While mentioning it in the introduction (line 52 onwards) is good, I don’t think it really needs to be stated again after stating the hypothesis here, which is to examine the reliability of sequence identification based on different database structures.

The construction of the databases (General DB, Restricted, Restricted-General) is not quite clear. If all relevant ITS2 sequences were downloaded for the General DB, why were so many discarded when clipped to the shorter amplicon target region? Was this mainly due to being low quality or did the General DB contain many sequences that do not (partially or to some extent) overlap the target region? Is the BLAST algorithm is searching against a database that does not contain quality or relevant sequences vs a “clean” database. It is quite common to use this “General DB” approach, but I think it should be explicitly stated that the General DB contains low-quality sequences as well as sequences from other regions in the gene outside the target amplicon.

I found it a bit difficult to understand the difference between Species origin and Database geographic area. Local species are the list of species recorded in Belgium, foreign species are everything else excluding local species. The Local DB are the sequences for the local species, while the World DB are sequences for Foreign + Local species (all species). While I understand why the authors wanted to explain the reason for including some cultivated species, I think that detail here makes it hard to follow the logic. “Local” species are basically a subset of a larger list of species, and could represent any number of scenarios. Overall, I think the tests done here are very applicable to pollen metabarcoding, but I did find it difficult to separate the detail around the study case from the actual question being addressed. For example, choosing two plant markers to test is a good idea, but in reality, any pair of markers of different lengths could have been selected.

Line 195: If a random sample is taken without replacement, does the remaining database get smaller after each consecutive time? Are you blasting against a smaller database at each iteration or replacing the sequences after BLASTing them?

Line 203 – “also we did not perform a full cross-validation of all the sequences” – does this mean that the randomly selected sequences you queried were restricted to local species and 10% of foreign species?


Results
Line 308: Can you summarise these results in a few words? It’s a complex figure and hard to remember what each label represents.

Line 322: As I understood it, Tophitplus is the same as TopHit except when there are ties for the top hit – how often did this happen?

Line 329: Are you saying that the accuracy is better when the actual sequence is present in the database (leaked CV) rather than when it is not (non-replacement CV)? When it is not present, is there another representative sequence of the same species?

Line 345: Is this because the species is no longer represented in the database once one sequence is taken out to run BLAST? Or do you think that BLAST struggles to find matches with shorter sequences?

Discussion
I think what would have really helped the reader here rather than just reporting the BLAST results, is to delve into why we see these different outcomes. For example, Line 488: “drops in taxonomic prediction accuracy probably because the number of sequences was dramatically lower in these restricted databases”. Since the whole point of the study is to test the performance of restricted databases, then I would expect the authors would be able to confirm this is the case.

---

## Round 0.2 · accepted · Accept

Reviewer #2 is mostly satisfied with your revisions. I think it's OK to go ahead with publication, but please have a look at this reviewer's re-review to see if it identifies any small textual changes you wish to do.

Reviewer 2 ·

Basic reporting

Manuscript meets these standards.

Experimental design

Clear explanations of the methods. The one area where I would like to see more detail is in the construction of the "restricted" databases - see additional comments below.

Validity of the findings

Manuscript meets these standards.

Additional comments

I appreciate the authors’ care in addressing the comments. The introduction, methods and conclusions are now much clearer to follow.

Additional comments:

Line 84: I would argue that BLAST is sequence similarity algorithm, rather than a classifier. The decision to assign taxonomy is made afterwards based on the taxonomic ID, bit score and threshold choice set by the researchers.

Line 86: BLAST provides a score of sequence similarity, not taxonomic assignment.

Line 88: I would add here something like “to assign taxonomy based on the similarity score produced by BLAST”

Line 96 – 110: I am not sure this paragraph is clear in its purpose. The three methods listed here serve different purposes. We can compare the accuracy of pollen metabarcoding in identifying species present in a sample to traditional methods such as microscopy (direct identification of biological samples). Metabarcoding of mock communities of known composition can tell us whether the metabarcoding process can accurately identify and quantify these particular communities. Cross-validation using BLAST can estimate the accuracy of the algorithm in matching a given sequence to particular reference databases. The preceding paragraph talks about parameter choice in pipelines, which is not relevant to microscopy.

Line 161: how did you build the restricted databases? Did you alter the primer binding parameters to ensure you captured as many sequences as possible? Did you test multiple alignments of sequences that were discarded in the restricted DB to check the primer binding sites? If the primers do not bind in-silico, would they bind in PCR?

Line 169: clipping the primer regions in the amplified sequences is a common step in metabarcoding analysis. Would doing this in-silico have helped with the searches?

Line 348: It would really be useful to know what is going on here. If you are blasting a reference sequence is no longer present in the restricted version of the database, why is it missing? Was it removed due to too many SNPs or poor quality? Consequently, is this reference sequence a realistic test if it is a poor-quality sequence or one that would not have been amplified because of the primer binding sites?

Line 360: “blast probably struggles”: can you provide a better explanation?

Line 374-375: Can you give more detail here (ie. what percentage of foreign flora is in local database, etc.)?

Line 472-473: The restricted databases only contained between 11-12% of the sequences from the full databases – this is a significant difference and it should be stated clearly.